# Capturing Longitudinal Changes in Brain Morphology Using Temporally Parameterized Neural Displacement Fields

**Aisha L. Shuaibu**                                                    A.SHUAIBU@SUSSEX.AC.UK
**Kieran A. Gibb**                                                        K.A.GIBB@SUSSEX.AC.UK
**Peter A. Wijeratne**                                              P.WIJERATNE@SUSSEX.AC.UK
**Ivor J.A. Simpson**                                                  I.SIMPSON@SUSSEX.AC.UK
*Department of Informatics, University of Sussex, Brighton, UK*

**Editors:** Accepted for publication at MIDL 2025

## Abstract

Longitudinal image registration enables studying temporal changes in brain morphology which is useful in applications where monitoring the growth or atrophy of specific structures is important. However this task is challenging due to; noise/artifacts in the data and quantifying small anatomical changes between sequential scans. We propose a novel longitudinal registration method that models structural changes using temporally parameterized neural displacement fields. Specifically, we implement an implicit neural representation (INR) using a multi-layer perceptron that serves as a continuous coordinate-based approximation of the deformation field *at any time point*. In effect, for any $N$ scans of a particular subject, our model takes as input a 3D spatial coordinate location $x, y, z$ and a corresponding temporal representation $t$ and learns to describe the continuous morphology of structures for both observed and unobserved points in time. Furthermore, we leverage the analytic derivatives of the INR to derive a new regularization function that enforces monotonic rate of change in the trajectory of the voxels, which is shown to provide more biologically plausible patterns. We demonstrate the effectiveness of our method on 4D brain MR registration. Our code is publicly available here.

**Keywords:** Longitudinal image registration, spatio-temporal regularization, monotonic regularization, implicit neural representations.

## 1. Introduction

Deformable image registration (DIR) plays an important role in medical imaging. The process involves obtaining a *spatially plausible* transformation that maximizes the similarity between a fixed and moving image pair. In longitudinal analysis (Hu et al., 2017; Durrleman et al., 2009, 2013), quantitative assessment of the changes in morphology over time is a key component in various applications such as; computational anatomy, population analysis, and disease diagnosis (Breijyeh and Karaman, 2020). For regions of pathological interest, these morphological features include the shape, volume, boundary, and extension to neighboring anatomical structures (Yang et al., 2020). Longitudinal image registration (LIR) quantifies subject-wise anatomical changes by aligning scans over multiple time points. However, LIR remains challenging due to several factors including; (a) image distortion, artifacts/noise present in the acquired data (Savitzky et al., 2018; Reuter et al., 2010), (b) LIR involves registering sequential scans with small anatomical variations which is difficult, (c) some methods rely on more sampled data per subject than is generally available (Qiu et al., 2009).

In this work, we propose a novel method for longitudinal image registration. Our key contributions include:

- A continuous implicit neural representation with periodic activation function that robustly models temporal and spatial transformations in longitudinal registration.

- A network that generates neural displacement fields (NDF) parameterized by time on a *continuous scale*, allowing interpolation between observed time points and extrapolation beyond the last observed data.

- We introduce a novel longitudinal regularization term and analyze its effect under different noise levels. We highlight the benefit of this derivation in inferring plausible longitudinal transforms.

## 2. Related Work

**Tensor-Based-Morphometry(TBM) for Neurodegenerative Diseases**: Statistical measures from tensor based morphometry (TBM) derived from the deformation fields between two registered images serve as efficient biomarkers used in disease progression analysis. A prominent application of LIR is in studying Alzheimer's Disease (AD) (Simpson et al., 2011; Qiu et al., 2008), a neurodegenerative progressive condition characterized by structural alterations in vulnerable regions of the brain such as the ventricles, hippocampus, and amygdala (Planche et al., 2022). AD biomarkers indicate this pathology unfolds as a continuous process over time (Dubois et al., 2016), and magnetic resonance imaging (MRI) modality is useful for characterizing these changes at a voxel level (Whitwell, 2009). The Jacobian determinant, $|J|$, is one of such TBM measures, interpreted as the expansion or contraction within a localized region -commonly used to assess the progression of AD-related atrophy (Chung et al., 2001). A $|J|$ value of 1 means no volume change, greater than 1 denotes expansion and between 0 and 1 is contraction of the local region.

**Image Registration:** Deep learning image registration methods that make use of convolutional neural networks (CNNs) have become ubiquitous (Bai and Hong, 2024; Wu et al., 2024; Balakrishnan et al., 2019), trained with moving and fixed image pairs, at test time they simply estimate the deformation field with a feed-forward pass. Despite their advantages these methods posses several constraints that diminish their practical applicability; (a) they require large training set to be able to generalize to unseen data (Wolterink et al., 2022), (b) they tend to under-perform on out-of-distribution data, (Vasiliuk et al., 2023), (c) we pay the cost of having a faster approximation of the deformation field with reduced accuracy (Hansen and Heinrich, 2021), (d) disease heterogeneity manifests differently across individuals (Han et al., 2020). LIR methods (Lee et al., 2023; Dong et al., 2021, 2024; Wu et al., 2024) often rely on these CNN-based architectures that maps between image intensities and a displacement field. Additionally, methods for pairwise image registration using tools like NiftyReg (Modat et al., 2014) or Advanced Normalization Tools (ANTs) (Avants et al., 2011) rely on hand-crafted optimization algorithms, unfortunately for longitudinal analysis these algorithms cannot easily generate displacement fields that are parameterized by time or work with scans of more than two time points per subject. This limits their practical ability to interpolate and extrapolate along points on the disease progression curve.

**Implicit Neural Representations:** Recent advancement has shown that fully connected neural networks serve as continuous, memory-efficient implicit representations used to model shape (Genova et al., 2019b,a), objects (Park et al., 2019; Atzmon and Lipman, 2020) or scenes (Gropp et al., 2020; Sitzmann et al., 2019). In image registration, INRs optimize a transformation function that maps each location $\mathbf{x}$ in one image to a location in another, this transformation is *implicitly* represented in the weights of an MLP and is of the form $\Phi(\mathbf{x}) = u(\mathbf{x}) + \mathbf{x}$, where $\mathbf{x}$ is the input coordinates and $u(\mathbf{x})$ is the predicted displacement field from the INR (Wolterink et al., 2022; Van Harten et al., 2023; Byra et al., 2023).

**Regularization in INRs:** Having realistic descriptions of the biological process in neurodegenerative diseases is critical in image analysis. For example, smoothness and monotonicity are commonly assumed when describing plausible evolution of the pathology of Alzheimer's disease (Abi Nader et al., 2020). Relying solely on surrogate measures such as image similarity is insufficient (Rohlfing, 2011), hence to avoid implausible deformations both spatially and temporally, regularization is generally enforced (Robinson et al., 2018; Metz et al., 2011; Yigitsoy et al., 2011; Durrleman et al., 2009; Fishbaugh et al., 2011). There is an advantage in the way INRs compute and represent displacement fields. Unlike other methods where the field is represented as a *fixed discrete grid* tied to the resolution of the fixed image, INRs define the field as a *continuous function over space* that maps any spatial coordinate to a displacement field. This means that displacement fields can be defined at arbitrary resolutions independent of the resolution of the fixed image. Given this representation of the displacement field and also the added advantage that neural networks are composed of differentiable operations, we can derive the analytic derivatives of the output transformation w.r.t the input coordinates. CNN-based methods (Balakrishnan et al., 2019; De Vos et al., 2019) often *approximate* this Jacobian matrix using finite difference and then minimize large gradients for regularization. With INRs, this matrix can be obtained directly, leading to more accurate gradients compared to numerical approximations (Wolterink et al., 2022). In our work, we leverage on this property of the INR to enforce regularization.

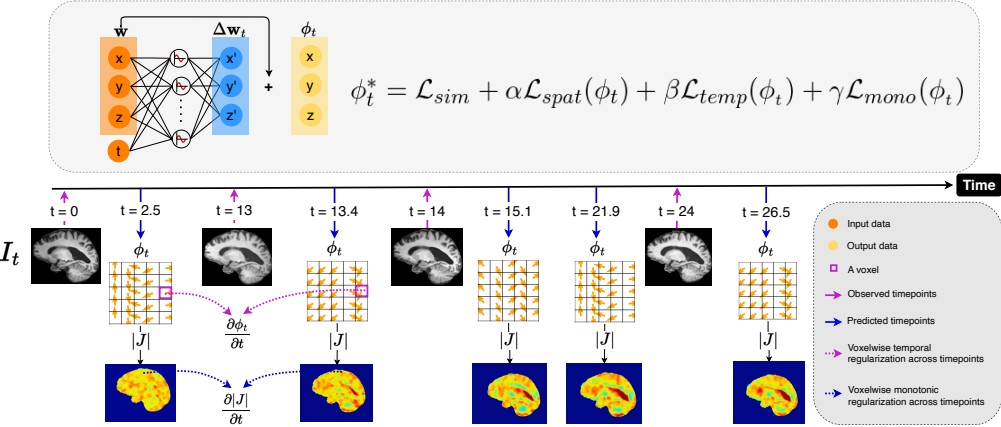

Figure 1: Given observed scans (pink arrows), our model represents the deformation field $\phi_t$ as a function (yellow circle). At inference, the model predicts time dependent fields, $\phi_t$ as well as $|J|$ maps for both observed and unobserved time points.

## 3. Method

Our proposed method is parameterized as follows; given 3D coordinates $\mathbf{w} = (x, y, z)$ and a corresponding time $t$ (since baseline) as input, we learn a function $h(\cdot)$ that produces displacement $\Delta \mathbf{w}_t$, describing the voxel-wise transformation at time $t$. The baseline image, defined as the earliest observed scan of a subject, is denoted as $I_0$ and subsequent *observed* follow-up scans represented as $I_t$. Our goal is to find a transformation $\phi_{\mathbf{t}} = \Delta \mathbf{w_t} + \mathbf{w}$ such that each location in an observed follow-up image, $I_t$ maps to a corresponding location on the baseline image $I_0$, *dependent on t*. A basic representation of this backward mapping is of the form: $\hat{I}_{t \to 0} = I_t \circ \phi_t$ and $\phi_t = Id$ when $t = 0$. The coordinates of the image domain are in the range $\mathbf{\Omega} \subset [-1, 1]$. Figure 1 illustrates the method.

### 3.1. Loss Function

Finding $\phi_t$ is formulated as an optimization problem where we seek to optimize the parameters $\psi$ of an MLP using gradient descent. The general formation of the loss function in longitudinal image registration is a combination of a similarity metric with spatial and temporal regularization. We introduce a third regularization term to the loss - the monotonic regularization. Hence the total loss is defined as;

$$\phi_t^* = \lambda \|\Delta \mathbf{w}_0\|^2 + \sum_{\mathbf{t} \in T} \mathcal{L}_{sim}(I_0, \hat{I}_{t \to 0}) + \alpha \mathcal{L}_{spat}(\phi_t) + \beta \mathcal{L}_{temp}(\phi_t) + \gamma \mathcal{L}_{mono}(\phi_t) \quad (1)$$

Where $\mathcal{L}_{sim}$ denotes normalized cross correlation between $I_0$ and $\hat{I}_{t \to 0}$ for $t > 0$. Since $\phi_{t=0} = Id$, we directly minimize the $L2$ norm at $t = 0$ by computing $\|\Delta \mathbf{w}_0\|^2$ of the displacements. Furthermore, we enforce spatially smooth deformations, $\mathcal{L}_{\text{spat}}(\phi_t)$ by penalizing large gradients of the analytically computed matrix defined as;

$$\mathcal{L}_{\text{spat}}(\phi_t) = \sum_{\mathbf{w} \in \Omega} \left( \frac{\partial \phi_t}{\partial \mathbf{w}} \right)^2, \quad \text{i.e.,} \quad \frac{\partial \phi_t}{\partial \mathbf{w}} = \begin{bmatrix} \frac{\partial (\phi_t)_x}{\partial x} & \frac{\partial (\phi_t)_x}{\partial y} & \frac{\partial (\phi_t)_x}{\partial z} \\ \frac{\partial (\phi_t)_y}{\partial x} & \frac{\partial (\phi_t)_y}{\partial y} & \frac{\partial (\phi_t)_y}{\partial z} \\ \frac{\partial (\phi_t)_z}{\partial x} & \frac{\partial (\phi_t)_z}{\partial y} & \frac{\partial (\phi_t)_z}{\partial z} \end{bmatrix} \quad (2)$$

To avoid unrealistic temporal deformations, we compute $\mathcal{L}_{temp}$ as the summation of individual analytic temporal derivatives of the displacement field expressed as;

$$\mathcal{L}_{\text{temp}}(\phi_t) = \sum_{t \in T} \sum_{\mathbf{w} \in \Omega} \left( \frac{\partial \phi_t(\mathbf{w})}{\partial t} \right)^2 \quad (3)$$

The analytic temporal derivative of the Jacobian determinant is a natural property derived from the INR and is accessible in the formulation, $\frac{\partial |J|}{\partial t}$. Biologically, we expect volumetric changes to progress monotonically over time (Abi Nader et al., 2020). This means that, at each voxel, the sign of $\frac{\partial |J|}{\partial t}$ should remain consistent over time, so voxel-wise volume change does not alternate between shrinking and growing. While therapeutic intervention may slow down progression, we do not expect a reverse in the trajectory of the disease (McColgan et al., 2023). Without explicitly enforcing this constraint, noise and other factors can introduce non-monotonic behavior in the model, disrupting the trajectory

of individual voxels. To address this, we propose a monotonic regularization term, $\mathcal{L}_{mono}$, that penalizes sign change in the temporal derivative of the Jacobian determinant. This loss term is expressed as:

$$\mathcal{L}_{\text{mono}}(\phi_t) = \min\left(\sum_t \max\left(\frac{\partial|J|}{\partial t}, 0\right), \sum_t \max\left(-\frac{\partial|J|}{\partial t}, 0\right)\right), \qquad (4)$$

Equation 4 defines the loss as the sum of all positive and negative contributions to that derivative, and we minimize the smaller of these contributions to encourage temporal consistency.

## 4. Experiments and Results

**Data and Implementation:** Adopted from (Sitzmann et al., 2020), the INR is represented as a function $h_\psi$ and is composed of five fully connected layers with sine activation between each layer. To embed temporal information, we build on architectures that combine scalar input with a primary network used for other tasks such as template construction (Dalca et al., 2019), segmentation (Kohl et al., 2018) and hyperparameter tuning (Hoopes et al., 2022). In our method, we extend the input to the registration function by introducing an auxiliary network $g_\theta$, that maps time to a feature space. This sub-network consists of a single hidden layer of 10 units and an output layer of 64 units with LeakyReLU as activation function. The output of $g_\theta$ is concatenated with each of the hidden layer representations from $h_\psi$ and used as input to the next layer. Preliminary studies evaluating different time embeddings is shown in Appendix A. We demonstrate our method using longitudinal 4D-T1 weighted MRI scans from ADNI (Petersen et al., 2010) that includes patients with Alzheimer's disease (AD), mild cognitive impairment (MCI) and healthy controls (CN). We select 10 from each group with 3 to 4 scans per subject aged between 65 to 89. For each subject, we perform affine registration from the follow-up scans to the corresponding baseline using NiftyReg (Modat et al., 2014), and extract anatomical labels with FreeSurfer (Fischl, 2012). The total time required to optimize the parameters of the MLP for a single subject (where gradients of all regularization terms are computed analytically) is approximately 90 minutes on an NVIDIA RTX A6000 GPU.

### 4.1. Experiment 1: Generating Temporally parameterized Neural displacement fields (NDF)

To evaluate the predictive ability of the proposed method, we assess its performance on both interpolation and extrapolation tasks. Specifically, we evaluate how well the model *interpolates between* two observed time points, and *extrapolates to future* time points. For this, we consider a subject with scans acquired at 0, 12, 24 and 37 months, representing different stages on the disease progression curve. In the interpolation experiment, we holdout the scan at $t = 24$ months and the model infers the displacement $\phi_{t=24}$ using the observations at 0, 12 and 37. Similarly, for the extrapolation experiment, we holdout the data at $t = 37$ months and the INR predicts time dependent displacement, $\phi_{t=37}$ with the available data points. We obtain Dice scores of 0.89 and 0.82 respectively. Figure 2 shows results from both cases and residual $|J|$ maps. Examples from other subjects is illustrated in Appendix C.

A key benefit of our method is its ability to predict $\phi_t$ at any given time point. In our experiments, we leverage this capability to encourage more plausible temporal deformations by adding temporal smoothness to the loss, even at unobserved time points.

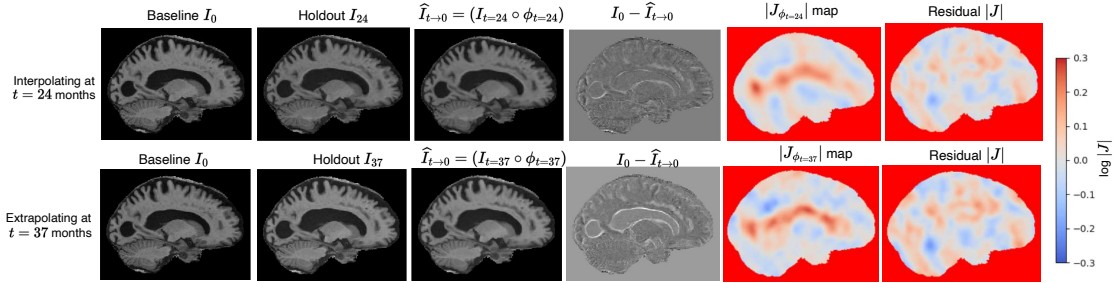

Figure 2: Interpolating and extrapolating Jacobian maps: The residual plot is the difference in $|J|$ maps when the time point is held out versus when it is used to fit the data. We provide examples from other subjects in Figure 14.

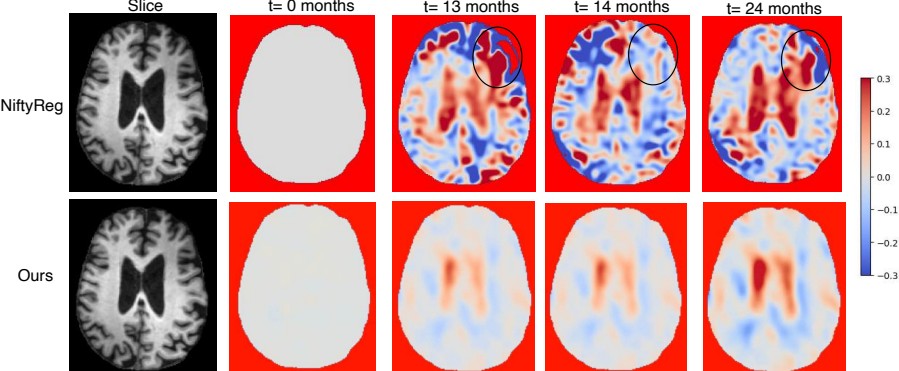

Figure 3: Comparing $|J|$ maps generated from NiftyReg with our proposed method. Black circle indicates inconsistent transformation over time. The bright red regions represents folded voxels.

Image registration is inherently an ill-posed problem, hence multiple plausible solutions exist. We conduct a sub-experiment comparing the $|J|$ maps generated by our method to those from NiftyReg (Modat et al., 2014) using an AD subject with observed scans at time points 0, 13, 14, and 24. Since NiftyReg only supports pairwise registration, we register each follow-up scan to the baseline image separately. The results are shown in Figure 3. The $|J|$ maps from NiftyReg shows transformations that are generally consistent with expected patterns in AD subjects, such as the expansion of ventricles. However, in certain regions (e.g., areas within the black circle), the transformations appear biologically implausible - expanding and then contracting over time, with folding of voxels occurring within the region. This is primarily because temporal dynamics in the longitudinal data is not captured.

## 4.2. Experiment 2: Analyzing the effect of the analytic temporal derivative of the Jacobian determinant w.r.t time - $\frac{\partial |J|}{\partial t}$

The monotonic loss defined in Equation 4 penalizes *voxel-wise* sign change over time. We conduct two experiments in this section that illustrates the effect of the proposed regularization. In the first experiment, we optimize our model on two instances; (a) using the proposed monotonic regularization, (b) without the regularization - Figure 11 in the Appendix B shows results from different combination of regularization terms; i.e spatial, temporal and monotonicity regularization and their effect on two metric scores.

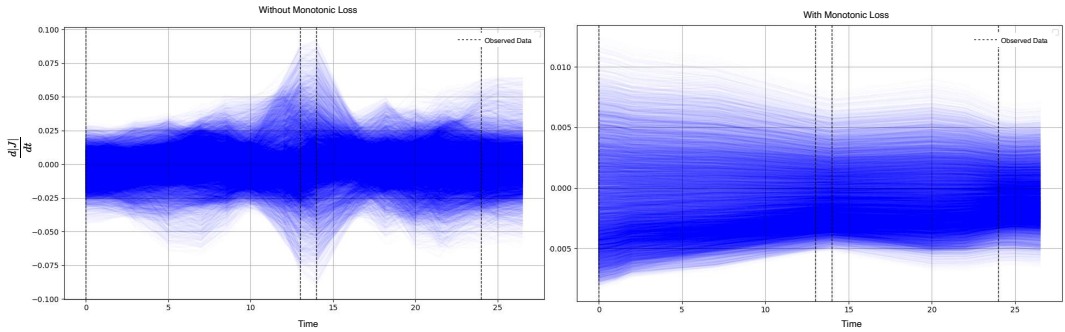

Figure 4: Voxel-wise rate of change within the Thalamus over a span of 26.5 months with and without the monotonic constraint. The black vertical lines are points where data was observed.

From the two instances above, we obtain the predicted $|J|$ maps at different time points over the span of two years and compute the voxel wise derivative over time on each structure. Figure 4 shows the results from both scenarios on the Thalamus of an AD patient. Without the monotonic loss, we observe erratic fluctuations and biologically implausible rate of change of approximately 2-5% per month in a span of two years. This is 5 times less with the monotonic constraint, where an overall rate of atrophy should be between 5-10% within two years (Hua et al., 2013). Figure 9 in Appendix B compares the transition of this rate of change on the ventricles in both scenarios. We also see the structure-wise effect as a violin plot in Figure 10.

Due to very little structural changes that LIR tasks aim to capture, small disruptions such as noise can have a huge effect on registration. In the second experiment, we evaluate the effect of monotonic loss on different noise levels. Noise drawn from a Gaussian distribution; $X \sim \mathcal{N}(\mu, \sigma)$ with a fixed mean of 0 and varying standard deviations of 0.15, 0.2, and 0.25 is added to the data. Each noise instance is ran 3 times and we plot the deviation to assess variability from random initialization. Figure 5 shows the effect of noise on the rate of change, $\frac{\partial |J|}{\partial t}$ and mean volume change, $|J|$ for three structures with and without monotonic regularization. With monotonic loss (right plot), at different noise levels the model is able to predict transformations that are consistent with expected patterns of the disease, however, this becomes more difficult as we increase the noise level. On the other hand, without monotonic loss, the result shows high sensitivity to noise and does not show biologically plausible pattern.

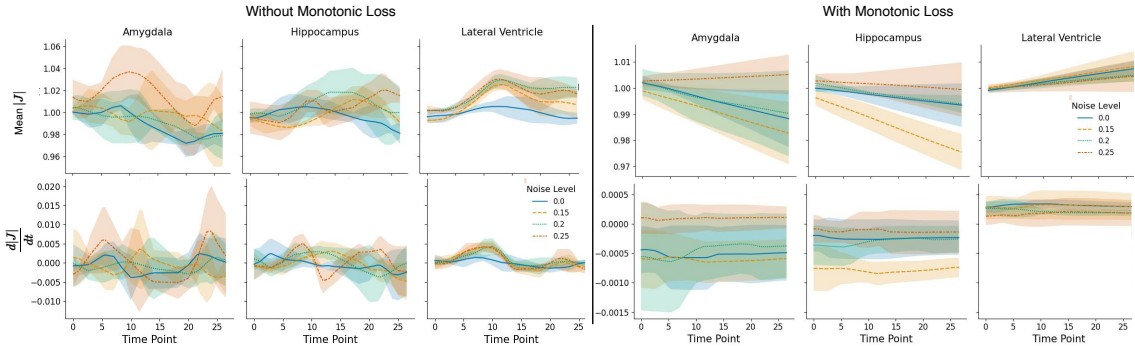

Figure 5: Effect of Noise; Top:mean $|J|$ plotted over 2 years, Bottom: $\frac{\partial |J|}{\partial t}$ for each structure.

### 4.3. Experiment 3: Progression comparison between AD, MCI and control

In this section, we conduct two experiments to show structural changes on subjects from the 3 groups. In the first experiment, we compare $|J|$ maps from a single subject per group, for visualization purpose, we provide results for just five time points in Figure 6 (left). The Figure shows an accelerated degeneration of anatomical structures (particularly the ventricles) for both AD and MCI patient while the control subject undergoes changes that seem more consistent with chronological aging. In the second experiment, we fit 10 different subjects per group to the INR (one INR for each subject) and compute the mean $|J|$ and $\frac{\partial |J|}{\partial t}$ to observe the trajectory of different structures over time across different groups. From Figure 6 (right), the mean $|J|$ plot (top) suggests an enlargement in the ventricle for both AD and MCI subjects (which aligns with expected patterns) and little to no structural changes for the control subjects. However, for the amygdala the average volume change decreases slightly for the MCI group and appears to have no volume change for AD subjects. Similarly, the hippocampus shows no average volume change for AD subjects and slight expansion for MCI subjects. Generally, we expect atrophy in both of these structures, (i.e., mean $|J|$ should decrease over time). The inconsistency may be due to several factors such as segmentation error, or the need for structure-wise optimal hyperparameter selection.

## 5. Discussion and Conclusion

We introduce a novel intra-subject LIR method that implicitly represents a transformation parameterized by time using an MLP with sine activation function. Our loss function is composed of three regularization terms; spatial temporal and monotonic regularization. The gradients in each of these terms are computed analytically, thus preventing approximation errors from methods like finite differences.

Our experiments demonstrates that our method is able to predict neural displacement fields and generate $|J|$ maps between and beyond observed time points. This may be useful in cases where scans are limited, to predict the morphology of structures over time. Furthermore, we introduce a novel regularization constraint, $\mathcal{L}_{mono}$, that enforces uni-directional trajectory of voxels over time - thus preventing biologically implausible transformations, and

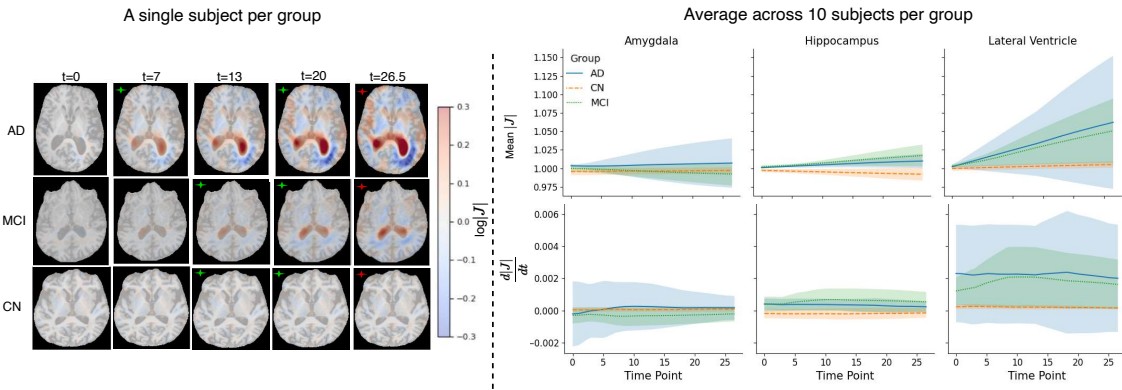

Figure 6: Comparing AD, MCI and control subjects. Left: A *subsample* of predicted $|J|$ maps overlayed on the baseline images for a single subject in each group. Red star represents extrapolated points, green represents *some* interpolated points and unstared frames are *some* observed scans for that subject. Right: $|J|$ metrics for 10 subjects in each group on different structures.

we evaluate its sensitivity to noise. Experiment 2 shows that by enforcing this monotonic constraint, we derive more biologically plausible rate of change that aligns more closely with the disease pathology (Hua et al., 2013).This work demonstrates an early stage version of our proposed method, hence predicting the voxel-wise volume change in each group (Figure 6) shows that our method has the *potential* to extract reliable volume change measurements that could serve as biomarkers for longitudinal studies to characterize neurodegenerative diseases (Wijeratne et al., 2023). Furthermore, unlike deep learning-based registration methods, ours does not rely on large training datasets; instead, it optimizes a separate network for each subject. While this may be computationally expensive, there are are several advantages of this method, including working with multiple time points with variable intervening periods. The comparison between Niftyreg and our method highlights the advantage of capturing temporal continuity in longitudinal analysis. Future work will look into amortized inference and a more efficient initialization method (van Harten et al., 2024) for computational efficiency. For small structures like the Amygdala and Hippocampus, our model struggles to accurately characterize the expected morphology in these regions, this may be due to errors in segmentation or hyperparameter tuning for that specific region- registration hyperparameters have a huge effect on the predicted deformation field (Hoopes et al., 2022; Shuaibu and Simpson, 2024; Mok and Chung, 2021). We will explore incorporating structure specific hyperparameter selection on the dataset. Finally, it would be interesting to investigate the effect of time parameterization on the main network. While preliminary studies in Appendix A shows that intermediate concatenation of time with the hidden layers in the main network broadly captures its effect. As an extension of this work, we plan to explore alternative parameterization methods such as leveraging a hypernetwork to generate the parameters of the main network (Ha et al., 2016). Where the input to the hypernetwork is a continuous time point.

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

# Appendix A. Integrating Time

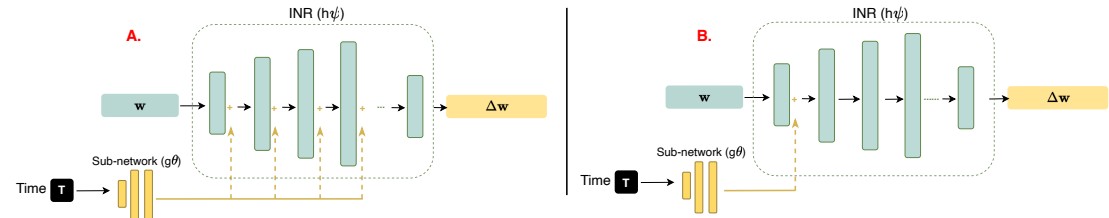

Figure 7: Time is parameterized by a sub-network, $g_\theta$, in the A (left) the output of $g$ is concatenated with each hidden layer of the primary network, $h_\psi$. In B (right), the time embedding from the sub-network is concatenated with the first hidden layer only. Results from both parameterization is shown below

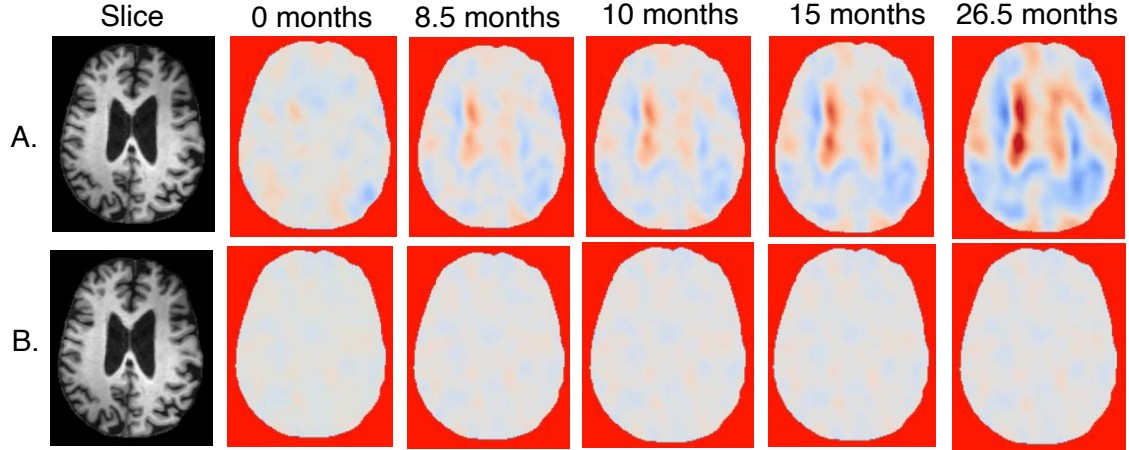

Figure 8: Results from parameterizing time using architectures A and B in Figure 6. The first row (A) is the result obtained when each hidden layer in the main network is concatenated with the time embedding (Figure 6A), the second row (B) is $|J|$ maps obtained when only the first hidden layer is concatenated with time embedding. While the first approach broadly captures the effect of the time embedding, the second one struggles to learn the effect of the embedding and outputs similar $|J|$ maps over time. Hence, all experiments carried out employ the first architecture.

# Appendix B. Experiments on Monotonicity

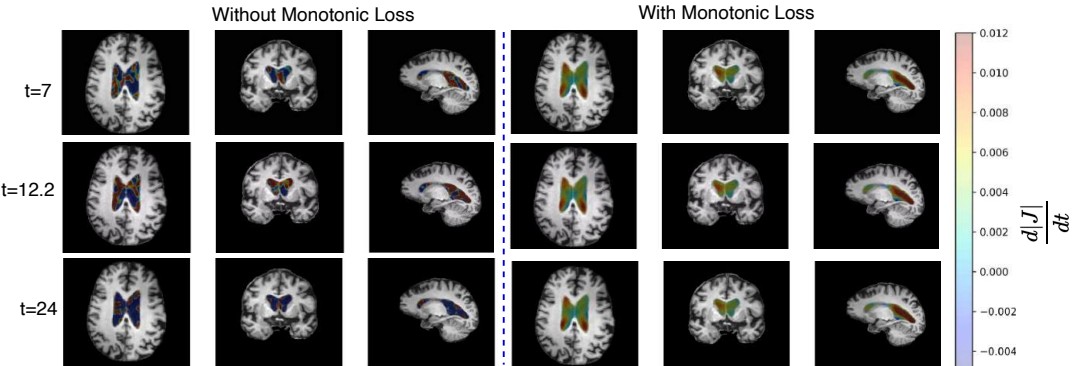

Figure 9: Comparing rate of change in the ventricles with and without monotonic constraint, we show results for 3 time points. Right: We observe a smoother transition on the rate of change with the added regularization. Left: Without this constraint, we observe discontinuities as a result of the erratic change of voxels trajectories over time.

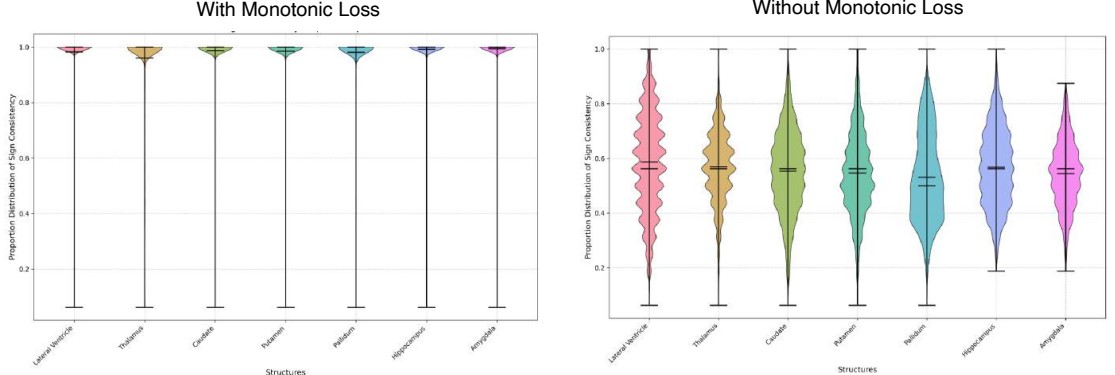

Figure 10: Proportion distribution of sign consistency for all voxels within a structure. We observe a multimodal distribution without monotonic regularization.

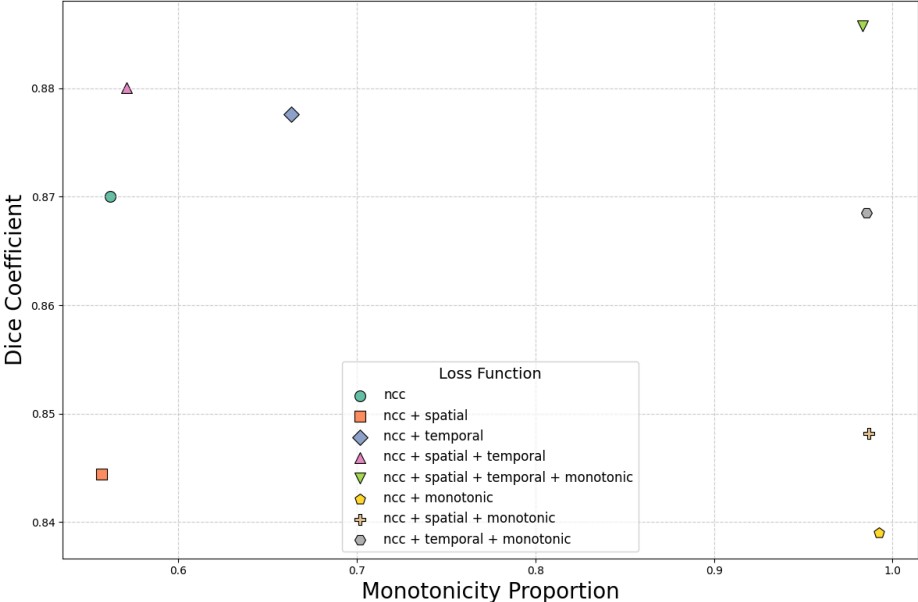

Figure 11: Selecting Optimal Regularization Combination: We measure the proportion of voxel-wise sign consistency within anatomical structures across time as a metric to quantify monotonicity. This figure shows results from different combination of regularization terms; i.e., spatial, temporal, and monotonicity regularization, and their impact on both the Dice score and proportion of monotonicity. To quantify the effectiveness of this constraint, we introduce a metric that provides assessment of how well the model adheres to the expected monotonicity in volume change, measured by the proportion of voxel-wise sign consistency within structures over time.

## Appendix C. Generating NDF's

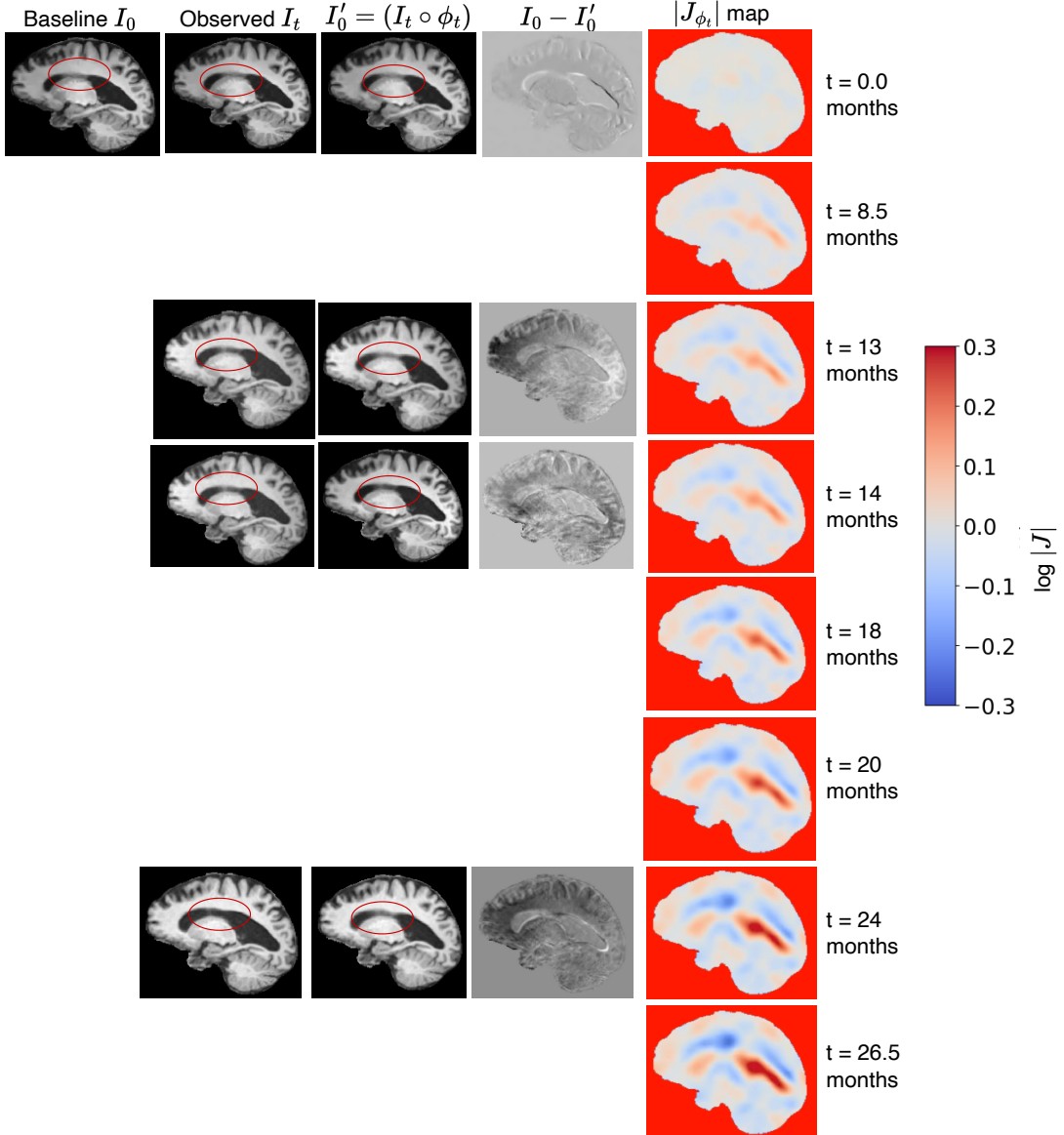

Figure 12: Predictive ability: Predicting $|J_{\phi_t}|$ maps at any time point.

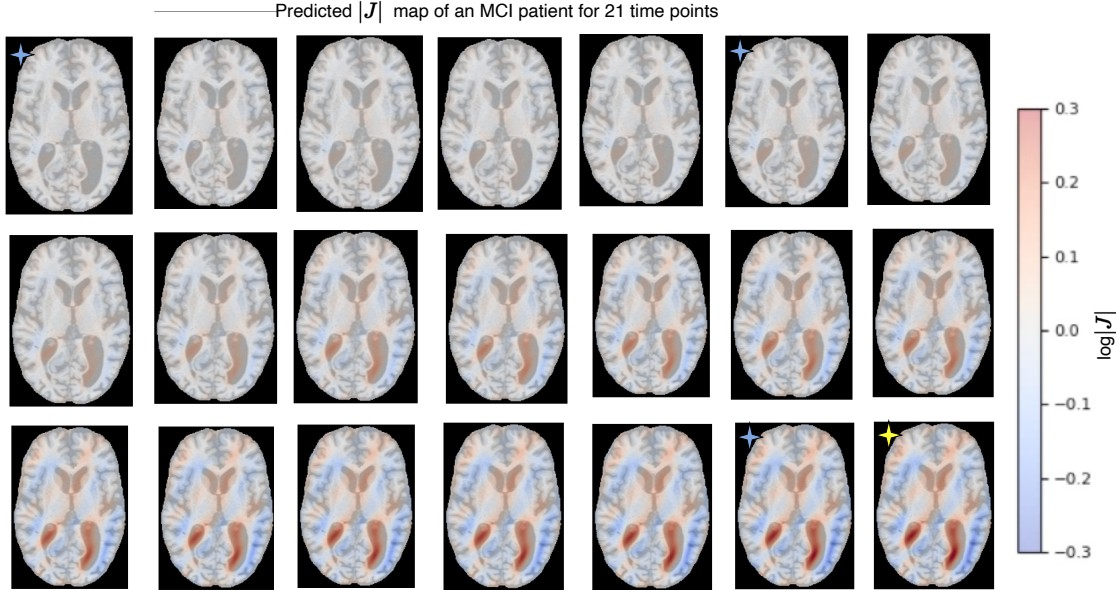

Figure 13: Predicted $|J|$ map for an MCI patient. The model observed only three scans. At time points 0, 12 and 24. The yellow star depicts extrapolated time point and blue stars are observed time points (the plot is interpreted from left to right, top to bottom).

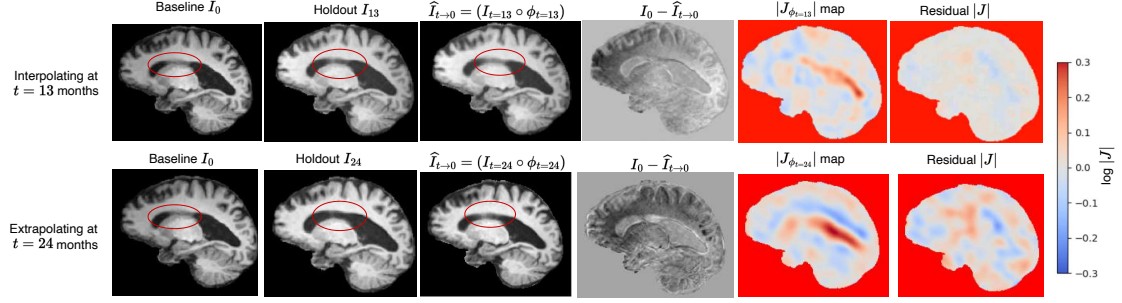

Figure 14: Other examples on interpolation and extrapolation.

