# OpenReview forum: "Capturing Longitudinal Changes in Brain Morphology Using Temporally Parameterized Neural Displacement Fields."
_MIDL.io/2025/Conference — MIDL 2025 Oral_

### Official Review · Reviewer_n5oj · 2025-02-18

**Confidence:** 4
**Preliminary Rating:** 5
**Recommendation:** Oral

**Summary:**

This paper introduces a new deep learning method to interpolate and extrapolate longitudinal brain images. They use a so-called implicit neural representation to model the displacement fields over time. The algorithm seems quite promising, in particular, the interesting constraint of monotonicity on the jacobian.

**Strengths:**

The paper is well-written, with a fairly complete review of previous methods, and a clear and concise explanation of the propose algorithm as well as a nice set of numerical experiments. The effect of the monotonic constraints is well-represented and quite interesting.

**Weaknesses:**

I did not find any major weaknesses in the paper to report here. As the author mentionned, this is a preliminary work that will be improved in future work, but I find it already quite good with many possible improvements.

**Detailed Comments:**

I don't have any detailed comments.

**Justification Of The Preliminary Rating:**

It is a good paper on an interesting topic and seems to work well, so it is worth discussing it during this conference. I recommend accepting it in the hope of reading the improved version next year .

**Questions To Address In The Rebuttal:**

I don't have any questions.

**Special Issue:**

No

---

> ### Author Response · Authors · 2025-03-08
>
> We thank the reviewer for taking their time to review our work and for the positive feedback.

---

### Official Review · Reviewer_7C6V · 2025-02-24

**Confidence:** 4
**Preliminary Rating:** 3
**Recommendation:** Poster
**Final Rating:** 4

**Summary:**

The authors introduce a novel longitudinal registration tool that models structural changes. The framework uses implicit neural representation in order to temporally parameterize the displacement fields. The novel contributions are: the use of a continuous implicit neural representation, an NDF-generating network parameterized by time and a new longitudinal regularization term.

Three sets of experiments are used to demonstrate the performance of the proposed framework.

**Strengths:**

The submission is well written.

The proposed method allows for both timepoint interpolation and extrapolation.

The proposed method introduces an assumption about monotonic volumetric changes to force temporal consistency. This eliminates a significant amount of noise

The experimental results are promising but somewhat limited. Figure 3, demonstrating the voxel-wise change over the thalamus, provides a good visual representation of the addition of the monotonic loss.

The code repository has been shared.

**Weaknesses:**

The experimental setup is small. No baseline evaluation from other methods, such as VoxelMorph (which was, for example, used as a reference in the intro).

The assumption about monotonic volumetric changes is very strong. It makes sense to eliminate noise, and it seems feasible to assume for aging; however, for studies with medical interventions (clinical trials), this might not hold.

Somewhat promising preliminary results. However, it is disappointing that the trends were not visible or contradictory for smaller structures. One would hope that by eliminating noisy predictions, these would be the areas that would benefit the most.

The authors decided to make the first timepoint as the baseline for the analysis. I believe that it might introduce bias into the analysis. Have they considered creating a template within the timepoints? (aka robust template creation closest to all points)

Affine registration is a pre-requisite.

**Detailed Comments:**

Did the authors use the longitudinal FreeSurfer pipeline for the image processing?

dice --> Dice

Fig 2: Am I reading this correctly that it is the cerebellar area whose extrapolation is the most challenging? Is this resolution-related or the anatomical variance in this area is thought to be the highest?

Can you change the yellow and white stars to other symbols/colors? On the already tiny figures distinguishing between these two colors/signs is extremely difficult.

**Justification Of The Final Rating:**

The reviewers answered many questions in a detailed way, made the description of the newly introduced method more clear. Even though I still have concerns about the eventual validation of the tool, the authors made updates to their discussion and experimental sections so I will increase my score to weak accept (poster).

**Justification Of The Preliminary Rating:**

The presented work introduces a promising framework, but the novelty level is moderate. The experimental results are preliminary and not yet fully convincing of the superiority of the method. A more complete description and discussion about some of the design choices are also missing from the current submission.

**Questions To Address In The Rebuttal:**

Can authors explain why they chose to make the first timepoint to be the baseline? Could the authors contemplate whether making the base computed in a more unbiased way help the prediction power?

A global alignment is a prerequisite. Could this also be learned? Or robustly done?

According to the authors some experimental inconsistency in the outcomes may be due to segmentation error, or the need for structure-wise optimal hyperparameter selection. How do they plan to test / validate these assumptions?

---

> ### Author Response · Authors · 2025-03-08
> **Response 1/2**
>
> We thank the reviewer for reviewing our work and the detailed feedback. Kindly note that revisions made to the paper are in red colour (however minor revisions are not annotated).\
> $\textbf{Weakness 1}$: The experimental setup is small. No baseline evaluation from other methods, such as VoxelMorph (which was, for example, used as a reference in the intro).\
> $\textbf{A}$: Thank you for this feedback. We acknowledge the benefits of comparing our work with an established longitudinal registration baseline but could not find a subject-wise longitudinal method that is parameterized by time , however we believe that altering methods like Voxelmorph to incorporate time complicates the model and is similar to proposing a new longitudinal DLIR method rather than a baseline to compare against. Could the reviewer kindly suggest or reference any other potential baseline methods you believe would be relevant for comparison? Additionally, we have included an extra experiment that compares against pairwise registration method. However these methods do not fully capture temporal dynamics in the longitudinal data. This is illustrated in Section 4.1 comparing $|J|$ generated from NiftyReg with those generated from our method. \
> $\textbf{Weakness 2}$: The assumption about monotonic volumetric changes is very strong. It makes sense to eliminate noise, and it seems feasible to assume for aging; however, for studies with medical interventions (clinical trials), this might not hold.\
> $\textbf{A}$: Thank you for your insightful comment. However, most Neurodegenerative diseases, including Huntington's, Alzheimer's, and Parkinson's, generally follow a monotonic progression [3]. In the case of a therapeutic intervention that affects the neurodegeneration we would expect it to either slow down or halt the progression and not to reverse it [2]. However, If the reviewer is aware of studies indicating a reversal of volumetric changes in response to interventions,  we would appreciate these being shared.\
> $\textbf{Weakness 3}$:Somewhat promising preliminary results. However, it is disappointing that the trends were not visible or contradictory for smaller structures. One would hope that by eliminating noisy predictions, these would be the areas that would benefit the most.\
> $\textbf{A}$: We agree with the reviewer. We believe that this is due to both segmentation error (from the segmentation method) and also the registration hyperparameter for those regions. Registration hyperparameters have a huge effect on the predicted deformation field. Currently, we use a fixed hyperparameter for all regularization terms across all structures,  while [4] and [5] shows that optimal hyperparameter varies per subject and brain region, as an extension to this work, we will explore structure specific optimal hyperparameter.\
> $\textbf{Weakness 4}$:The authors decided to make the first timepoint as the baseline for the analysis. I believe that it might introduce bias into the analysis. Have they considered creating a template within the timepoints? (aka robust template creation closest to all points)\
> $\textbf{A}$: Thank you for this insightful comment. An important consideration is that to describe temporal changes, we would ideally prefer  $|J|$ to reflect changes wrt a specific, known time point rather than an abstract, unknown time point, which would be the case for an estimated template (since all follow up scans need to be mapped to the space of a known time point). However, we will explore ways to reduce the bias to the baseline in future work.\
> $\textbf{Weakness 5}$: Affine registration is a pre-requisite.\
> $\textbf{A}$: Yes, Agreed. We do this using NiftyReg - highlighted in Section 4.\
> $\textbf{Detailed Comments}:$ \
> $\textbf{Comment 1}:$ Did the authors use the longitudinal FreeSurfer pipeline for the image processing?\
> $\textbf{A}$: We used NifytyReg for preprocessing and extracted labels using FreeSurfer.\
> $\textbf{Comment 2}:$ dice --> Dice\
> $\textbf{A}$: Thank you for pointing this out, it has been corrected and highlighted in red.\
> $\textbf{Comment 3}:$ Fig 2: Am I reading this correctly that it is the cerebellar area whose extrapolation is the most challenging? Is this resolution-related or the anatomical variance in this area is thought to be the highest?\
> $\textbf{A}:$ Thank you for pointing this out (although we had to change the subject used for that experiment due to recommendations from reviewer 1). However, to incorporate this suggestion, since our method has the ability to predict deformation fields at arbitrary time points (including unobserved points), we leverage this capability by adding temporal smoothness to the loss, even at unobserved time points to improve extrapolation capability.\
> $\textbf{Comment 4}:$ Can you change the yellow and white stars to other symbols/colors? On the already tiny figures distinguishing between these two colors/signs is extremely difficult.\
> $\textbf{A}:$ Adjusted to a different colour. Thank you\

---

> > ### Author Response · Authors · 2025-03-08
> > **Response 2/2**
> >
> > $\textbf{Comment 5}:$ A global alignment is a prerequisite. Could this also be learned? Or robustly done?\
> > $\textbf{A}:$ We do global alignment as a preprocessing step using Niftyreg [1], however this is something that can also be inferred. We have noted this suggestion as a potential experiment to look at in future work.\
> > $\textbf{Comment 6}:$According to the authors some experimental inconsistency in the outcomes may be due to segmentation error, or the need for structure-wise optimal hyperparameter selection. How do they plan to test / validate these assumptions?\
> > $\textbf{A}:$Thank you for this comment. Automatic segmentation is a process that is generally associated with variance [6]. To test/validate this, we would need manually segmented longitudinal data for the subjects, which is generally difficult to acquire.\
> > $\textbf{References}:$
> > 1. Marc Modat, David M Cash, Pankaj Daga, Gavin P Winston, John S Duncan, and Sébastien Ourselin. Global image registration using a symmetric block-matching approach. Journal of medical imaging, 1(2):024003–024003, 2014\
> > 2. McColgan, P., Thobhani, A., Boak, L., Schobel, S. A., Nicotra, A., Palermo, G., ... & GENERATION HD1 Investigators. (2023). Tominersen in adults with manifest Huntington's disease. The New England journal of medicine, 389(23), 2203-2205.\
> > 3. Abi Nader, C., Ayache, N., Robert, P., Lorenzi, M., & Alzheimer’s Disease Neuroimaging Initiative. (2020). Monotonic Gaussian Process for spatio-temporal disease progression modeling in brain imaging data. Neuroimage, 205, 116266\
> > 4. Hoopes, A., Hoffmann, M., Fischl, B., Guttag, J., & Dalca, A. V. (2021). Hypermorph: Amortized hyperparameter learning for image registration. In Information Processing in Medical Imaging: 27th International Conference, IPMI 2021, Virtual Event, June 28–June 30, 2021, Proceedings 27 (pp. 3-17). Springer International Publishing.\
> > 5. Shuaibu, A. L., & Simpson, I. J. (2024). HyperPredict: Estimating Hyperparameter Effects for Instance-Specific Regularization in Deformable Image Registration. arXiv preprint arXiv:2403.02069.\
> > 6. Joskowicz, L., Cohen, D., Caplan, N., & Sosna, J. (2018). Automatic segmentation variability estimation with segmentation priors. Medical image analysis, 50, 54-64.

---

> > > ### Comment · Reviewer_7C6V · 2025-03-12
> > >
> > > Thanks for the detailed responses (to both my and the other reviewers' comments)
> > >
> > > * even if subject-wise longitudinal methods do not exist, selecting baselines tools is still informative; tools that can handle modality and subject differences, could have an easier time with intra-subject examples
> > >
> > > * longitudinal freesurfer -- this is not a longitudinal registration method, but FreeSurfer segmentations are already done, so the longitudinal pipeline should be used here. Explicitly using the longitudinal nature of the acquisitions several disease-related studies have been carried out and delta changes can be quantified looking at segmentation stats at the end
> > >
> > > * additionally the robust registration tool used with the longitudinal pipeline (with affine alignment, by default) might serve as an ideal baseline, esp when small changes are expected
> > >
> > > * Fig 2; new figure does not have the same issue as before, but it is still unclear why the cerebellar region differences were so pronounced in the previous example.
> > >
> > > * re comm 6: The authors do not really state whether there are any plans / possibility to test their proposed hypothesis.
> > >
> > > * re new Fig 3: The ground truth must lie between the two presented solutions. The proposed outcome presents quite subdued changes compared to NiftyReg. Is there only change expected in the ventricles?

---

> > ### Author Response · Authors · 2025-03-14
> > **Response 2/2**
> >
> > $\textbf{Comment 6: }$ re new Fig 3: The ground truth must lie between the two presented solutions. The proposed outcome presents quite subdued changes compared to NiftyReg. Is there only change expected in the ventricles?
> >
> > $\textbf{A: }$ Thanks for this comment. Registration however is  an ill posed problem, hence no ground truth transformation exists. Regarding the subdued changes, we also expect changes in other regions (i.e blue in the figure)- but the ventricles are large structures hence the change is more prominent. Furthermore, regularization has a huge effect on registration tasks and selecting what the optimal value should be is a problem on its own (kindly refer to  point in response 1 about selecting appropriate baselines). In Essence, It is not that the true solution lies in the middle, it's more that there are a variety of answers that explain the data and we’re trying to find a solution that is biologically plausible and gives us useful information about diseases.
> >
> > $\textbf{References: }$
> > Wang, G., Li, W., Aertsen, M., Deprest, J., Ourselin, S., & Vercauteren, T. (2019). Aleatoric uncertainty estimation with test-time augmentation for medical image segmentation with convolutional neural networks. Neurocomputing, 338, 34-45.

---

> ### Author Response · Authors · 2025-03-14
> **Response 1/2**
>
> We thank the reviewer once again for their insightful feedback. Below are the responses to the comments made.
>
> $\textbf{Comment 1: }$ even if subject-wise longitudinal methods do not exist, selecting baselines tools is still informative; tools that can handle modality and subject differences, could have an easier time with intra-subject examples.
>
> $\textbf{A: }$ Yes, we agree that a variety of registration tools exist. However, Inter-subject differences differ  from intra-subject differences in that the latter is sensitive to smaller changes. As a result, there are several design choices that need to be made in terms of level and form of regularization, similarity metric and other important hyperparameters. All of these have to be optimized to give a fair comparison with any baseline method - by default they are not suited to intra-subject registration due to the small-scale nature of the changes. For instance, Niftyreg (the example figure in the paper) uses Normalized Mutual information as a cost function which is robust to modality changes, it also uses bending energy as  regularization with a default weight of 0.005. Selecting the level of regularization and the form is important for specific tasks.  While we have not conducted extensive heuristics, we have provided a comparison with NiftyReg as a reference. A comprehensive evaluation across multiple tools and settings would be a significant undertaking, which we leave for future work.
>
> $\textbf{Comment 2: }$ longitudinal freesurfer -- this is not a longitudinal registration method, but FreeSurfer segmentations are already done, so the longitudinal pipeline should be used here. Explicitly using the longitudinal nature of the acquisitions several disease-related studies have been carried out and delta changes can be quantified looking at segmentation stats at the end.
>
> $\textbf{A: }$ We agree with the reviewer, we currently use the default Fresurefer which may not be optimal for longitudinal data. Due to the current  time constraint, we reserve the longitudinal FreeSurfer pipeline for future iterations of this work. Although, one thing to note is that using the longitudinal pipeline approach, true changes in the segmentations might be lost due to the bias in the segmentation tool (i.e., since we have segmented the images together,  the segmentations will be similar including propagation of segmentation error across all segmentation). Hence for the purpose of evaluation,  looking at local shape changes may become difficult using the generated segmentations. However, we will look into this further and do a comparison between both approaches in our extended work. Thank you for pointing this out.
>
> $\textbf{Comment 3: }$ additionally the robust registration tool used with the longitudinal pipeline (with affine alignment, by default) might serve as an ideal baseline, esp when small changes are expected
>
> $\textbf{A: }$ Thanks for this comment. However, to the best of our knowledge, the longitudinal freesurfer pipeline only uses an Affine transformation - this wouldn't be able to describe local shape changes(non-linear changes) which is what we are interested in.
>
>
> $\textbf{Comment 4: }$ Fig 2; new figure does not have the same issue as before, but it is still unclear why the cerebellar region differences were so pronounced in the previous example.
>
>
> $\textbf{A: }$ Thanks for pointing this out. As previously mentioned  we had to look at a new subject for that experiment due to the comments from reviewer 1. However, we realised that part of the reasons the difference was pronounced was due to the time embedding - parameterized as a nonlinear function of time. We found that after the last observed time point, extrapolation may be challenging (which the previous figure was showing). Hence, to aid extrapolation we compute temporal regularization even at unobserved time points this helps to ensure a smoother temporal transformation over time. We have included updated results from the previous subject in Figure 14 Appendix C.
>
>
> $\textbf{Comment 5: }$ re comm 6: The authors do not really state whether there are any plans / possibility to test their proposed hypothesis.
>
> $\textbf{A: }$ We could look at an ensemble of segmentation tools  to try and reduce the variance, and see what effect this has and also methods like [1]. Another approach is to acquire  manual segmentations, however we do not have facilities for manual segmentations for this dataset. These are the two ways we can address this however they are outside the  scope of this paper, hence we reserve this for future work.

---

### Official Review · Reviewer_p3rp · 2025-02-24

**Confidence:** 3
**Preliminary Rating:** 3
**Recommendation:** Poster
**Final Rating:** 4

**Summary:**

This paper presents a novel longitudinal image registration (LIR) framework using Neural Displacement Fields (NDFs). The method leverages Implicit Neural Representations (INRs) for brain MRI data. A key innovation is the introduction of a time-dependent deformation field, allowing interpolation and extrapolation of anatomical changes between observed and unobserved time points. The proposed approach is evaluated on longitudinal 4D-T1 MRI scans from the ADNI dataset, demonstrating its ability to track neurodegenerative changes in Alzheimer’s Disease (AD), Mild Cognitive Impairment (MCI), and control subjects. Experimental results suggest that the method can generate smooth displacement fields and maintain temporal consistency.

**Strengths:**

- The use of Implicit Neural Representations (INRs) for time-dependent registration is a novel idea in the context of longitudinal neuroimaging. The method’s ability to interpolate and extrapolate structural changes over time provides an advantage over conventional pairwise registration approaches, which are limited to discrete time points.
- The authors conduct experiments on longitudinal MRI data from the ADNI dataset, which is a well-established benchmark for studying Alzheimer’s disease progression. The study includes interpolation, extrapolation, and disease progression analysis, providing a thorough evaluation of the method’s capabilities.

**Weaknesses:**

- The method is compared primarily against itself (with and without monotonic regularization) rather than against established longitudinal registration baselines.
- Related to this, it would be beneficial to clearly define the key contributions and emphasize them more explicitly in both the methodology and experimental sections. Additionally, the design and role of the auxiliary temporal network are not clearly explained. Providing more rationale behind the architectural choices—particularly how the temporal auxiliary network influences the INR model—would strengthen the clarity and impact of the proposed approach.
- The paper does not discuss the computational cost of training and inference. How does the method compare to traditional registration techniques in terms of runtime and memory usage? INRs use rather small models but must be trained in each subject. And this can really influence the clinical impact of registration methods.

**Detailed Comments:**

Major comments
- The output of the temporal auxiliary network is concatenated with each hidden layer of the INR network, but the reasoning behind this architectural design is not entirely clear. Could the authors provide more intuition for this choice and clarify its impact on performance?
- In Experiment 1, the chosen timestamps for evaluation are 0, 13, 14, and 24 months, with 13 months selected for interpolation. However, 13 and 14 months are very close in time, raising concerns. Would it be possible to evaluate interpolation at a more distant time point, or to provide results across multiple interpolation settings for a more robust assessment?
- A significant portion of the results are qualitative or based on a small number of subjects. To better demonstrate the effectiveness of the approach, it would be beneficial to include more quantitative evaluations or to statistically validate the findings (e.g., statistical significance testing, confidence intervals, or comparisons against baseline methods).
- It is unclear how the method would perform in cases of large structural changes, such as the appearance of new anatomical structures or when tracking subjects undergoing a change in diagnosis (e.g., MCI to AD). Could the authors discuss how the model handles such cases, particularly when sudden morphological changes occur?

Minor comments:
- Some additional results are included in the appendix but are not clearly referenced in the main text. While not all details need to be repeated, a brief mention of key findings in the main body would improve clarity and guide readers to relevant supplementary materials.

**Justification Of The Final Rating:**

The authors have provided a detailed and thorough response to the concerns raised, along with relevant edits that improve the clarity of the contributions, related work, and experimental evaluation. The revisions make the methodology and results more transparent, strengthening the overall presentation.

These improvements justify a higher rating, and I am increasing my score to 4: Weak Accept.

**Justification Of The Preliminary Rating:**

While the paper is well-written and well-organized, and the proposed method appears interesting and promising, several key weaknesses outweigh its strengths. In particular, some methodological choices and experimental settings lack clarity, making it difficult to fully assess the approach. Additionally, the absence of comparisons with existing methods significantly limits the ability to gauge the true relevance and impact of the proposed framework.
Given these concerns, I recommend a preliminary rating of 3: Borderline.

**Questions To Address In The Rebuttal:**

- Address the concerns raised in the weaknesses section, particularly regarding the justification of claims, methodology clarity, and evaluation details.
- Improve the clarity of the manuscript by incorporating the points highlighted in the weaknesses and detailed comments, ensuring better explanation and presentation across the relevant sections.

---

> ### Author Response · Authors · 2025-03-08
> **Response 1/2**
>
> We thank the reviewer for taking their time to go though our work, we have tried to address all the concerns raised. Kindly note that revisions made to the paper is in red colour (however minor revisions like referencing figures are not annotated in red).\
> $\textbf{Weakness 1}$: The method is compared primarily against itself (with and without monotonic regularization) rather than against established longitudinal registration baselines.\
> $\textbf{A}$:Thank you for this feedback, we acknowledge the benefits of comparing our work with an established longitudinal registration baseline that accommodates multiple timepoints. We are aware of  general DLIR methods like Voxelmorph [6] and Longformer [7] but could not find a subject-wise longitudinal method that is parameterized by time. We believe that altering the above mentioned methods introduces complexities - making it similar to proposing a brand new registration method. Could the reviewer kindly suggest or reference any potential baseline methods they believe would be relevant for comparison? We conducted an additional experiment to compare against pairwise registration methods like NiftyReg however these methods do not fully capture temporal dynamics in the longitudinal data - illustrated in Section 4.1. \
> $\textbf{Weakness 2}$: Related to this, it would be beneficial to clearly define the key contributions and emphasize them more explicitly in both the methodology and experimental sections. Additionally, the design and role of the auxiliary temporal network are not clearly explained. Providing more rationale behind the architectural choices—particularly how the temporal auxiliary network influences the INR model—would strengthen the clarity and impact of the proposed approach.\
> $\textbf{A}$: Thank you for this comment, we have restated the contributions in the methodology section (particularly sections 3.0.1 and 4.0) and also extended the discussion section. For the comment about time embedding, we have added useful references in Section 4 (where the idea came from). Furthermore, we agree that the time embedding needs to be explored further. We conducted a preliminary experiment where we explore other forms of time embedding, we show the results for this in Appendix A. As an extension of this work, we plan to explore alternative parameterization methods such as leveraging a hypernetwork to generate the parameters of the main network [1].\
> $\textbf{Weakness 3}$: The paper does not discuss the computational cost of training and inference. How does the method compare to traditional registration techniques in terms of runtime and memory usage? INRs use rather small models but must be trained in each subject. And this can really influence the clinical impact of registration methods.\
> $\textbf{A}$:Thanks for pointing this out, we have included in brief the computational cost in Section 4. Yes, it can be computationally expensive since we optimise a single model per subject. The total time required to optimize the parameters of the MLP for a subject is approximately 90 minutes on an NVIDIA RTX A6000 GPU - also considering that gradients are computed analytically for all three regularization terms, the computational cost is a similar to that reported in [8]. However there are several advantages of this method such as working with multiple time points with variable intervening periods . Another advantage of optimizing subject-wise rather than training a DLIR model is that we do not need to rely on the availability of large training dataset, and also disease heterogeneity manifests differently across individuals [5], this method has the ability to capture that. Future work will look into amortised inference, or at least efficient initialisation [4] for computational efficiency - however this may be complicated due to varying number of images for each subject.\
> $\textbf{Detailed Comments:}$\
> $\textbf{comment 1:}$ The output of the temporal auxiliary network is concatenated with each hidden layer of the INR network, but the reasoning behind this architectural design is not entirely clear. Could the authors provide more intuition for this choice and clarify its impact on performance?\
> $\textbf{A:}$ Thank you for pointing this out, addressed in answer to weakness 2\
> $\textbf{comment 2:}$ In Experiment 1, the chosen timestamps for evaluation are 0, 13, 14, and 24 months, with 13 months selected for interpolation. However, 13 and 14 months are very close in time, raising concerns. Would it be possible to evaluate interpolation at a more distant time point, or to provide results across multiple interpolation settings for a more robust assessment?\
> $\textbf{A:}$ We have changed to a different subject with longer time intervals for this experiment, updated in Section 4.1, other examples are in Appendix C
>
> $\textbf{Response is split due to character limit- all references are in the second section below}$

---

> ### Author Response · Authors · 2025-03-08
> **Response 2/2**
>
> $\textbf{comment 3:}$ A significant portion of the results are qualitative or based on a small number of subjects. To better demonstrate the effectiveness of the approach, it would be beneficial to include more quantitative evaluations or to statistically validate the findings (e.g., statistical significance testing, confidence intervals, or comparisons against baseline methods).\
> $\textbf{A:}$ Thank you for pointing this out. The current study demonstrates our method’s capabilities and emphasises the contributions mainly through qualitative results,  as mentioned by the reviewer. We acknowledge that large scale quantitative analysis would be beneficial to fully elucidate the benefits of this work, but this is a substantial undertaking with limited time. We have noted this as an avenue for future work.\
> $\textbf{comment 4:}$It is unclear how the method would perform in cases of large structural changes, such as the appearance of new anatomical structures or when tracking subjects undergoing a change in diagnosis (e.g., MCI to AD). Could the authors discuss how the model handles such cases, particularly when sudden morphological changes occur?\
> $\textbf{A:}$ We are working with intra-subject longitudinal registration for neurodegenerative diseases where the time scale is fairly small. This means that generally we expect reasonably small, monotonic, morphological changes (even in the case of disease) in the structures over time, mentioned in Section 1 and highlighted in [2] [3]. In general, we expect much smaller deformations will be required compared to inter-subject image registration. Furthermore, we do not expect new anatomical structures to appear and disappear over time, however we expect a change in the shape of the anatomical structures over time - which is what we are characterising. If the reviewer means the appearance of  brain lesions over time, we acknowledge that this is possible and is a general problem for any registration method/tool. We have also included the references above to the main paper- Section 3.0.1.\
> $\textbf{comment 5:}$ Some additional results are included in the appendix but are not clearly referenced in the main text. While not all details need to be repeated, a brief mention of key findings in the main body would improve clarity and guide readers to relevant supplementary materials.\
> $\textbf{A:}$ We agree with the reviewer and have adjusted references  in the main text (mostly in experiment and results section) to guide readers to the Appendix (kindly note that not all is highlighted in red as they are not major changes).\
> $\textbf{References:}$
> 1. Ha, D., Dai, A., & Le, Q. V. (2016). Hypernetworks. arXiv preprint arXiv:1609.09106.\
> 2. Bône, A., Colliot, O., & Durrleman, S. (2018). Learning distributions of shape trajectories from longitudinal datasets: a hierarchical model on a manifold of diffeomorphisms. In Proceedings of the IEEE conference on computer vision and pattern recognition (pp. 9271-9280).\
> 3. Hua, X., Hibar, D. P., Ching, C. R., Boyle, C. P., Rajagopalan, P., Gutman, B. A., ... & Alzheimer's Disease Neuroimaging Initiative. (2013). Unbiased tensor-based morphometry: improved robustness and sample size estimates for Alzheimer's disease clinical trials. Neuroimage, 66, 648-661.\
> 4. van Harten, L., Van Herten, R. L. M., & Isgum, I. (2024, December). REINDIR: Repeated Embedding Infusion for Neural Deformable Image Registration. In Medical Imaging with Deep Learning.\
> 5. Han, Xu, et al. "A deep network for joint registration and reconstruction of images with pathologies." Machine Learning in Medical Imaging: 11th International Workshop, MLMI 2020, Held in Conjunction with MICCAI 2020, Lima, Peru, October 4, 2020, Proceedings 11. Springer International Publishing, 2020.\
> 6. Balakrishnan, G., Zhao, A., Sabuncu, M. R., Guttag, J., & Dalca, A. V. (2019). Voxelmorph: a learning framework for deformable medical image registration. IEEE transactions on medical imaging, 38(8), 1788-1800.\
> 7. Chen, Q., Fu, Q., Bai, H., & Hong, Y. (2024). Longformer: longitudinal transformer for Alzheimer's disease classification with structural MRIs. In Proceedings of the IEEE/CVF winter conference on applications of computer vision (pp. 3575-3584).\
> 8. Wolterink, J. M., Zwienenberg, J. C., & Brune, C. (2022, December). Implicit neural representations for deformable image registration. In International Conference on medical imaging with deep learning (pp. 1349-1359). PMLR.

---

### Comment · Area_Chair_hyQ5 · 2025-02-24
**Missing review**

The missing review will be submitted today (24 Feb).

---

### Author Rebuttal · Authors · 2025-03-08

**Rebuttal:**

We thank the reviewers for taking their time to provide valuable feedback. However, we noticed references were not attached to the comments. It would be helpful to provide specific references to guide us to the relevant literature.

**Supporting Material:**

/attachment/f90a7c3217c32b3079ffc2d341bca2babc10fa89.pdf

---

### Meta-Review · Area_Chair_hyQ5 · 2025-03-20

**Recommendation:** Accept (Poster)
**Confidence:** 5

**Metareview:**

The paper describes the use of time-conditioned implicit neural representations for longitudinal image registration. It is well-written and well-prepared and includes extensive experiments on the ADNI dataset. Longitudinal image registration is relevant for many applications. The authors engaged well in the discussion and resolved some of the issues mentioned by the reviewers in their initial review.

One of the weaknesses highlighted by the reviewers is the limited comparison to existing methods. Moreover, as is often the case with implicit neural representations, reviewers question the algorithm's runtime. Furthermore, while the authors assert that their method extrapolates over time, I wonder about the practical value of these extrapolations. Since the method is purely model-driven and no generalization occurs between patients (unlike 'conventional' deep learning approaches), the same extrapolation could be achieved through simple linear extrapolation.

Nevertheless, this work will likely be of interest to the general MIDL community. Its strengths outweigh its weaknesses and include a novel regularization term and interesting experiments.